# Prioritizing public health? Factors affecting the issuance of stay-at-home orders in response to COVID-19 in Africa

**Gregg R. Murray**[1]*, **Joshua Rutland**[2]

**1** Department of Social Sciences, Political Science, Center for Bioethics and Health Policy, Augusta University, Augusta, Georgia, United States of America, **2** Department of Social Sciences, MAISS, Augusta University, Augusta, Georgia, United States of America

* gmurray@augusta.edu

**Data Availability Statement:** The dataset generated and analyzed for this study are available on OSF at https://osf.io/xmq7u/ (DOI 10.17605/OSF.IO/XMQ7U).

## Abstract

COVID-19 has sickened and killed millions of people globally. Conventional non-pharmaceutical interventions, particularly stay-at-home orders (SAHOs), though effective for limiting the spread of disease have significantly disrupted social and economic systems. The effects also have been dramatic in Africa, where many states are already vulnerable due to their developmental status. This study is designed to test hypotheses derived from the public health policymaking literature regarding the roles played by medical and political factors as well as social, economic, and external factors in African countries' issuance of SAHOs in response to the early stages of the COVID-19 pandemic. Using event history analysis, this study analyzed these five common factors related to public health policy to determine their impact on African states' varying decisions regarding the issuance of SAHOs. The results of this analysis suggest that medical factors significantly influenced decisions as did factors external to the states, while the role of political factors was limited. Social and economic factors played no discernible role. Overall, this study suggests how African leaders prioritized competing factors in the early stages of a public health crisis.

## Introduction

COVID-19 has disrupted medical, social, and economic systems worldwide. Rampant infections and high death counts have led governments to implement several policies aimed at containing and combating the disease. These efforts have ranged from the relatively benign to the oppressive, from public awareness campaigns, curfews, and social distancing orders to limits on public gatherings, travel restrictions, and school and business closures. In many cases, governments have resorted to stay-at-home orders (SAHOs), which have restricted people to only essential activities such as meeting medical and sustenance needs [1]. Public health policies such as SAHOs that tightly restrict social interaction have been shown to effectively limit the spread of the disease [2], but the consequences in non-health domains have been severe [3]. School closures have impacted over 60 percent of all students globally, with localized closures affecting millions more [4]. Business closures have caused major economic ripples across a

**Funding:** The authors received no specific funding for this work.

**Competing interests:** The authors declare no competing interests.

wide range of countries [5], leaving millions furloughed [6] and predictions that the pandemic may result in an additional 70 to 100 million people living in extreme poverty into the foreseeable future [7]. Because of disruptive effects like these, decisions to issue SAHOs are not taken lightly by government leaders.

The impacts on Africa have been substantial. The continent surpassed more than 100,000 deaths from COVID-19 within the first year of the pandemic [8]. Public opinion surveys of African Union member states conducted about six months into the pandemic indicate that seven in 10 respondents say they had experienced difficulties getting food and a similar proportion reported lower income than the previous year [9]. Further, the effects of the pandemic are expected to lead to a drop in GDP across African countries of between 1.4% and 7.8% [10]. Despite substantial variation in economic and social conditions, African countries are exclusively categorized as "developing economies" in the UN's 2020 World Economic Situation and Prospects report [11], and many struggle to provide basic social services. The pandemic exacerbates this already challenging situation, particularly in Sub-Saharan Africa.

The public health situation in Africa and elsewhere created by the coronavirus is dire, which makes SAHOs, one of public health policy makers' most effective tools for fighting infectious disease, an obvious policy option. But the serious and far-reaching social and economic consequences of SAHOs are highly predictable and far from trivial, which also make SAHOs a very costly policy option for governments to take. The objective of this study is to assess the relative roles of these competing considerations. In particular, it is to test hypotheses derived from the public health policymaking literature regarding the roles played by medical and political factors, which have been identified as key forces in public health policy [12, 13], as well as social, economic, and external factors in motivating governments in Africa to issue a SAHO in response to COVID-19 (N = 41) or not (N = 13) in the first five months of the pandemic. To the authors' knowledge, this is the first study to comprehensively evaluate the role that the five common factors related to public health policy played in the issuance of SAHOs in response to COVID-19 in Africa.

Given this objective, the following section describes the policy context of Africa, and the next presents previous research that provides theoretical explanations for African government's responses to the pandemic along with nine associated hypotheses to be tested. This review focuses on common medical, political, social, economic, and external factors that often influence public health policy creation [14]. The next section details the method of analysis used, event history analysis, and the data collected, which come from sources such as the World Bank, United Nations, World Health Organization (WHO), and Oxford University. The results section offers several analyses and robustness checks of the findings, which, ultimately, suggest that African governments' decisions were significantly influenced by medical factors and factors external to the countries, only questionably influenced by political factors, and not influenced at all by social or economic factors. Finally, the study concludes by summarizing the results, addressing this study's limitations, and placing its findings in a larger context about pandemic-motivated decision making.

## The African policy context

With more than 50 countries, over 3,000 ethnicities, and between 1,250 and 2,100 spoken languages, Africa is one of the most culturally and politically complex regions on Earth (e.g., [15]). Societies across the continent function with widely varying institutional structures and governing ideologies that are influenced by colonial legacies, tribal heritage, and long-standing ethnic fractures (e.g., [16, 17]). Due to later democratization, political parties have been constrained by limited experience and, as a result, political institutions have remained under-

developed in many cases [18]. Many African governments have been crippled for decades by corruption, ineffective policy creation, and terroristic armed militias. Wars, coups, and ethnic massacres have characterized the volatile political atmosphere and highlighted the problems caused by poorly considered borders inherited from previous colonial powers [16, 17].

However, not all of Africa confronts the same problems or struggles equally. The northern region is frequently studied in conjunction with the Middle East due to their greater cultural, political, and economic similarities [19–21]. Northern Africa is comprised of predominantly Arab states including Algeria, Egypt, Libya, Morocco, South Sudan, Sudan, and Tunisia. Tunisia is an outlier in the region as the only state to successfully transition to a democratic regime following the Arab Spring in 2011 [22]. The rest of Northern Africa's authoritarian government structures, pro-Middle Eastern foreign policies, and political and economic alliances, like OPEC, reflect a deep connection with the Middle East [23–25]. In particular, oil production gives North African countries greater economic resources than many others on the continent.

The rest of the continent, referred to commonly as Sub-Saharan Africa, can be divided into four sub-regions according to the United Nations Statistical Division: Eastern, Middle, Western, and Southern. Each of these sub-regions has distinct cultures and policy contexts, drawing both from their own diverse histories and the influence of former colonial powers such as Britain and France [18]. Large states such as Nigeria possess significant developmental potential but have been subject to disconnects between abundant resources and policy direction [26]. In several African states, the aftermath of independence saw shifts away from political pluralism and the consolidation of power under one political party [18]. This has resulted in the retention of authoritarian rule for states like Libya while some states, such as Ghana, have seen eventual democratic reform attempts with varying levels of success. Other states like the Central African Republic have suffered a series of coups and civil wars that installed rulers with varying interests in democracy [18, 27, 28]. As these situations demonstrate, Africa is far from a single conglomerate of culturally or politically homogenous countries and, according to some, even dividing Africa into Northern and Sub-Saharan regions fails to capture its true diversity [26].

The vulnerable political nature of Africa is of even greater concern in the context of a pandemic. Failure to mitigate a disease outbreak can cause lost confidence in the state, resulting in the loss of domestic and even global legitimacy in extreme cases [29]. This can trigger internal and external political instability, degrading national security even in developed states [29, 30]. While this danger may make African states more likely to act to mitigate the pandemic, failure may result in a spike in regional instability and conflict. The resulting refugee crises, for instance, would place even greater strain on a region troubled by multiple long-standing refugee situations while also facilitating more rapid transmission of the virus across borders as infected people flee to safety [29]. Failure against a pandemic can exacerbate long-term humanitarian crises.

Public health and development remain salient topics in African politics and economics. The continent carries a disproportionate share of the global disease burden [31] and of epidemics [32] relative to its population. Health crises like AIDS, malaria, cholera, influenza, tuberculosis, and Ebola have ravaged the continent and demonstrated the consequences of inadequately structured healthcare policies (e.g., [33]). The United Nations' International Health Regulations (IHR) offer a benchmark for evaluating states' preparedness to mitigate pandemics. The IHR identifies "core capacities," or baseline requirements, for factors deemed essential to combating pandemic diseases [34]. These capacities include "national legislation, policy and financing, coordination and National Focal Point communications, surveillance, response, preparedness, risk communication, and human resources and laboratories" [34]. In 2019, Africa averaged 46% prepared for pandemics based on the IHR core capacities scores,

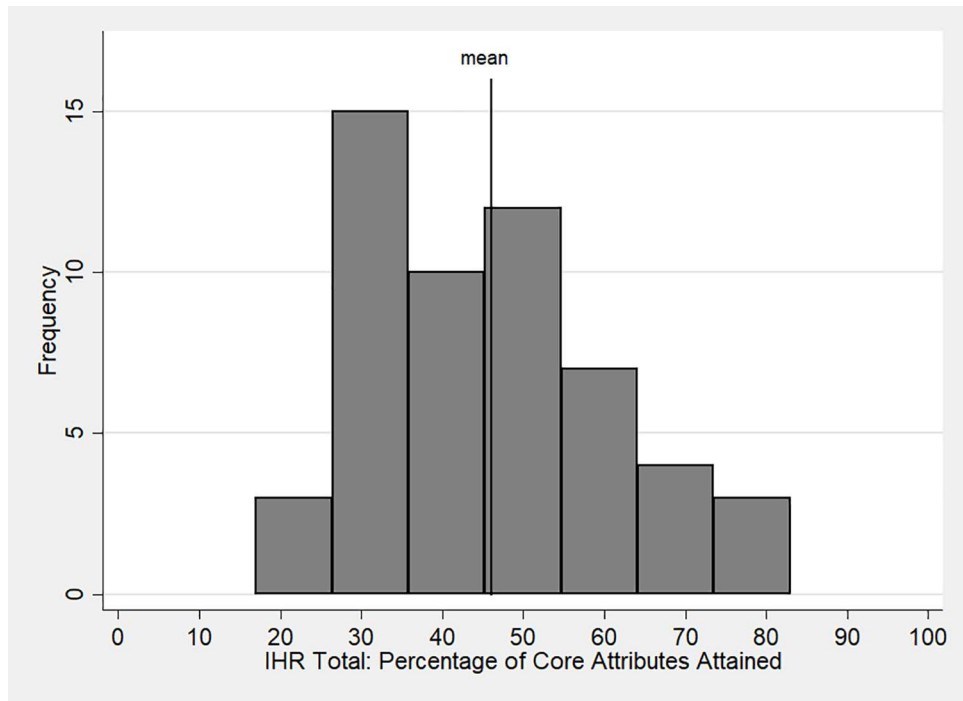

**Fig 1. Distribution of UN International Health Regulations core capacities in Africa.**

while countries globally averaged 63% [35]. As Fig 1 shows, though, the average masks a great deal of variation in Africa. Scores ranged from lows of 25% or less in the Central African Republic, Equatorial Guinea, Guinea-Bissau, and Lesotho to highs of 70% or more in Algeria, Egypt, Rwanda, South Africa, and Tunisia.

While AIDS had and continues to have a profound effect in Africa, the response to Ebola seems more pertinent to the response to COVID-19 due to its recency and means of transmission. Ebola impacted three West African nations most significantly: Guinea, Liberia, and Sierra Leone. These states suffered significant loss of life throughout the two-year outbreak, culminating in 11,000 deaths and inflicting significant impacts on the countries' healthcare systems and economies [36]. Being three of the poorest countries in the world, under-funding and under-preparedness of their health systems contributed to Ebola's ability to manifest as a pandemic [37]. For instance, the three countries lacked many of the conventional programs of strong public health systems such as effective prevention, vaccination, monitoring, isolation, and health promotion strategies. Moreover, they lacked enough healthcare workers in general and specialist healthcare workers in particular [38]. Recent measures show their ongoing public health predicament. They are at or slightly below the African mean in IHR total scores with Sierra Leon having attained 40% of core capacities for surveillance and response to public health events of international concern, Guinea having attained 44%, and Liberia 46%. These scores, though, put the countries well below the global mean of 63%. But during the Ebola crisis the countries deployed what resources they had to acquire tests, isolate people who tested positive, and trace and quarantine others who may have come in contact with the disease [38]. Neighboring states such as Mali and Kenya began their response strategies with border closures and bans on air travel to and from the epicenters [37]. In all, though, the Ebola crisis highlighted the weaknesses of African public health systems [37]. The hospital-based curative care structure existing in the post-colonial era was insufficient and under-equipped to control

the outbreak efficiently, particularly in countries and on a continent already bearing a dispro-portionately high communicable disease burden compared to the rest of the world [37]. Even-tually, foreign intervention helped end the crisis despite an initially lethargic response on the part of the WHO [36, 37]. Western countries, particularly France, Great Britain, and the United States, sent thousands of troops and healthcare workers and pushed aggressively to develop treatments and potential vaccines.

## Literature and hypotheses

With the policy context of Africa in mind, one can begin to connect theory to the practice of responses to the COVID-19 pandemic. One way to conceptualize policymaking is by focusing on internal and external determinants of policy adoption [39, 40]. Generally, internal determi-nants focus on political, social, and economic characteristics specific to each country that might affect that country's adoption of a policy. Because science- and medicine-related policies are highly complex and require specialized assessment [41], health policy researchers often add scientific or medical characteristics to the list of internal determinants [14]. External deter-minants, which are often conceptualized as "diffusion," focus on the actions of other countries that may affect a country's adoption of a policy. In particular, policies are thought to diffuse across countries geographically and temporally (e.g., [42, 43]). The following overview of the literature provides a framework for developing a number of hypotheses related to internal and external factors expected to affect the issuance of COVID-19-related SAHOs in Africa.

## Internal factors

**Medical factors.**　A popular refrain is to "follow the science" in regard to government responses to the COVID-19 pandemic (e.g., [44]). It makes sense, then, that the physical threat of a serious disease should hold some sway over leaders' decisions to implement SAHOs. But beyond common sense, theory suggests and research shows that salient issues, that is, issues that are important to a nontrivial proportion of a population, are more likely to rise on the political agenda and be addressed more quickly [41, 45]. This is particularly the case in crisis situations where "life-sustaining functions of a social system" are confronted by transbound-ary, unique, and uncertain threats [46]. More specifically, the rapid spread of a dangerous dis-ease may capture the public's attention as well as render a country's healthcare system incapable of meeting the demand for staff, equipment, and other medical resources necessary to care for patients. Taking steps to reduce a disease's transmission rate and reduce the likeli-hood of overwhelming a country's healthcare system is referred to in epidemiology as "flatten-ing the curve" [47, 48]. SAHOs are regulations designed to force social distancing and reduce disease transmission to help flatten the epidemiological curve. Leaders of countries where the disease is more prevalent may face greater public scrutiny and be more concerned about the capacity of their healthcare system to handle the increase in cases and, therefore, may be more aggressive in combating the virus by issuing a SAHO. This suggests:

*H1a*: *Governments will be more likely to issue a SAHO as the threat of the disease increases.*

Arguments and findings from policy research suggest that countries learn from their own experiences (e.g., [49]; see also [46]). This suggests that experience with the outbreak of the lethal Ebola virus in Africa between 2014 and 2016 may also play a role in states' decisions to issue SAHOs. West Africa, in particular Guinea, Liberia, and Sierra Leone were hit hard by the virus, which killed more than 70% of the roughly 20,000 people it infected within the first nine months of the outbreak [50]. At around the nine-month mark, the World Health Organization (WHO) indicated that a failure to implement more drastic control measures would result in

"an increase from hundreds to thousands" of new cases and deaths in the future [50]. Many of the measures recommended for preventing the spread of Ebola are similar to those recommended to slow the spread of the COVID-19, and reports indicate that the three countries that experienced the worst Ebola outbreaks implemented many of the same procedures and used many of the same tools in their efforts against COVID-19 [38]. Altogether this suggests:

*H1b. Governments that experienced recent Ebola outbreaks will be more likely to issue a SAHO.*

While countries can desire to adopt certain policies, in many cases the number of feasible policies may be limited due to a variety of factors, including resource constraints [51]. Healthcare systems vary dramatically in quality [52], and this is certainly true regarding preparation for the COVID-19 pandemic [53]. States with healthcare systems that are not fully prepared for a widespread infectious disease may have to depend more on non-pharmaceutical interventions, such as SAHOs, to protect citizens. This suggests:

*H1c. Governments with healthcare systems that are less prepared for a pandemic will be more likely to issue a SAHO.*

**Political factors.** By definition, public policy is the product of governmental processes. Some researchers have argued that political factors play the key role in public health policy [13]. Thinking about the roles of public health considerations relative to other considerations, Oliver [12] concluded, "Politics, for better or worse, plays a critical role in health affairs." Among the political forces, research suggests that government structure plays a significant role in the difficulty and complexity involved in policymaking. Concentration of power is a major dimension in government structure, and policy should be easier to make in governments with more centralized powers as there are fewer actors from whom to gain approval (e.g., [54]). In particular, policymaking is more centralized in authoritarian regimes with fewer decision makers, while the reverse is true in democratic regimes. This suggests a decision to issue a disruptive policy may be easier to make in an authoritarian regime. From another perspective, democratic governments tend to promote citizen freedoms and choices (e.g., [55]), while authoritarian governments limit them (e.g., [56]). This, again, indicates that authoritarian regimes may be more likely to limit citizens' choices by issuing a SAHO than democratic regimes. Both perspectives suggest:

*H2a: Authoritarian governments will be more likely to issue a SAHO than democratic governments.*

A country's governance capacity should also play an important role in policymaking, particularly during a crisis (e.g., [46]). A government with greater governance capacity—e.g., coordination, delivery, regulation, and analysis—has access to a larger number of administrative and policy tools and greater ability to use them. Moreover, these governments are often perceived as more legitimate by the public further enhancing their ability to make and implement policy [46]. On the other hand, failure to implement adopted policies undermines government and leader legitimacy [57]. Authorities of fragile or ineffective countries may avoid adopting disruptive policies such as SAHO's out of the realization that they cannot enforce them and may, therefore, simply undermine themselves with a SAHO. This suggests:

*H2b: Governments with greater administrative capacity will be more likely to issue a SAHO.*

Colonial legacies in Africa are manifest in modern government structures, borders, and social cleavages [58]. Most African countries structured their political institutions in a manner similar to their colonizers', though the results have varied [18]. As such, former colonizing

countries may have affected the formation of and expectations for healthcare policies in African states [36]. Evidence suggests that France exercised more direct control over its colonies in Africa, which influenced the adoption of more centralized power structures in former French colonies. The British, though, often established satellite governments to handle their colonies' administrative needs [59]. Much like the argument regarding concentration of power and regime type, this suggests:

H2c: *Governments with a French colonial history will be more likely to issue a SAHO than governments with a British colonial history.*

**Social context.** The social context of a country and its citizens also plays a consequential role in public health policy [14]. Populations with different cultural and historic characteristics can take different approaches to policy [36]. One issue is society's orientation toward collectivism versus individualism. In collectivist cultures people tend to think of themselves as part of a group and prioritize group over individual goals, while in individualist cultures they tend to think of themselves as separate from a group and prioritize individual over group goals [60]. This difference is manifested in public health policy by the type of healthcare system a country employs. Blank, Burau, and Kuhlmann [36] broadly categorize healthcare systems in terms of healthcare provision based on national health service, social insurance, or privatized insurance. These categories reflect systems' institutional characteristics, which are usually arrayed along a continuum from government monopoly to free market. National health services, which include universal healthcare coverage financed by taxes, exist at one end of the continuum and are consistent with the notion that healthcare is viewed more as a public good or collective responsibility. The opposite end of the continuum is characterized by private insurance, which consists of individual responsibility for healthcare acquisition financed by private or employer contributions and the view that healthcare is more of a private good or individual responsibility. A government with a monopoly on healthcare services would be more likely to issue a SAHO out of a sense of collective responsibility to its citizens, while free market systems might be less likely due to their greater focus on individual responsibility for healthcare. This suggests:

H3a: *Governments in states where healthcare is viewed as a public good are more likely to issue a SAHO than governments in states where healthcare is viewed as a private responsibility.*

Sociodemographic characteristics also affect government policymaking. Vulnerability to disease varies, for instance, by age, sex, ethnicity, education, income, and marital status (e.g., [61]). As such, government policies vary to account for variations in the populations they serve. In terms of policy regarding infectious diseases, large numbers of people living and working in close proximity to one another have a greater likelihood of physical contact with disease carriers and, therefore, of spreading such diseases. This suggests urban populations are especially vulnerable to the spread of infectious diseases [62]. It is likely, then, that the populations of some countries are more vulnerable than the populations of other countries. In these situations, it is reasonable to expect that government leaders will act more aggressively to address the disease. This suggests:

H3b: *Governments will be more likely to issue a SAHO if their population is more vulnerable to infectious disease due to socio-demographic characteristics such as urbanicity, age, education, or ethnic composition.*

**Economic context.** Research suggests that higher gross domestic product (GDP) tends to be indicative of greater overall wealth, which has been identified as an important factor in

countries' healthcare policies [63]. In particular, the provision of healthcare is expensive, so it is not surprising that evidence indicates poorer countries are less able to fund healthcare systems and that wealthier countries spend two to three times more on healthcare than poorer countries (e.g., [64]). As such, poorer countries may be less able to care for sick patients and more willing to absorb the social and economic costs of a lockdown. This suggests:

*H4*: *Governments with weaker economies will be more likely to issue a SAHO.*

## External factors: Diffusion

Policymaking in a jurisdiction is also often affected by influences outside that jurisdiction. For instance, policies "diffuse" from one country to another because of interdependencies between countries (e.g., [65, 66]), and healthcare policy is no exception [67]. Policy researchers have identified several mechanisms for government-to-government policy diffusion: learning, normative pressure, competition, and coercion (e.g., [49]). These mechanisms operate through several pathways such as structural equivalence in a network and joint membership in a group [66], but researchers widely view geographic proximity as the fundamental approach to policy diffusion [68]. Geographic diffusion suggests that policies spread to geographically close or neighboring states particularly because of the effects of shared or similar socio-demographics, cultural values, and media (e.g., [69]). Further, the possibility for a disease to spread physically across jurisdictional borders makes geographic proximity a particularly salient consideration during a pandemic [2, 29]. This suggests that:

*H5a*: *A government will be more likely to issue a SAHO when more governments of bordering countries have issued a SAHO.*

The diffusion of policies, like other innovations [42], is temporal as well as geographic [43]. In general, innovations can diffuse following a number of patterns or shapes over time, including a linear pattern of continuous increase and an immediate effect followed by no change over time [70]. That said, substantial evidence indicates that innovations have long diffused across time following a sigmoidal or S-shaped curve that reflects a social learning pattern of slow growth followed by accelerating and then decelerating growth to saturation [42]. For instance, Cistercian monasteries temporally diffused across Europe in the 1100s following this pattern as did resistance to the introduction of mechanical threshing in rural England in the 1830s. In the US, the growth of the canal, railway, telegraph, road, and oil pipeline networks followed a similar pattern in the 1800s and 1900s [42]. Not surprisingly, as public policies are innovations, much contemporary policy research finds a similar temporal diffusion pattern [40]. As Nicholson-Crotty [45] concludes, "One of the key insights from the large literature [on the diffusion of public policies among governments] is that policies typically diffuse temporally in a relatively consistent and predictable manner. More specifically, it suggests that the diffusion process produces an S-shaped cumulative frequency distribution." For example, Mooney and Lee [43] noted this pattern in the adoption of abortion regulation reform in US states in the 1960s and 1970s, while Murray and Jilani-Hyler [21] found the same diffusion pattern in COVID-19-related SAHOs in the Middle East and North Africa region. In particular, researchers explain this temporal pattern as a process of social learning in which states learn from policy results of similar states such that a small number of regional leaders implement a new policy from which other states observe, learn, and adopt at an accelerating then decelerating rate. In the case of COVID-19, the urgent and uncertain nature of the pandemic [46] in the early months likely compressed the policy process rendering longer-term trends like health policy transition in Africa [71] less influential [45]. Overall, this suggests:

*H5b*: *The probability a government will issue a SAHO will accelerate then decelerate over time as a response to other governments issuing a SAHO.*

## Method and data

Policy researchers often use panel data and event history analysis (EHA) to identify factors related to the probability that a policy will be adopted during a given time period [39, 40]. The panel data constitute the "event history," which in this case captures every day from January 31 to June 7, 2020, for each of the 54 African countries with observations in the form of country-days. It is important that the timeframe under consideration is theoretically significant [72]. January 31 is the day after the WHO declared an international emergency related to COVID-19, and June 7 is 30 days after the last African country issued a SAHO during the first wave of the pandemic. The decision to terminate the event history 30 days after the last issuance is based on a number of considerations. None of the countries that refrained from issuing a SAHO to that point later issued a SAHO in 2020. One month is a discrete and conventional timeframe. It balances the need to give reasonable time to capture additional issuances (the mean [median] time between issuances was 2.7 [1] days and the maximum was 18 days) with the need to avoid confounding the effects of time by selecting an overextended timeframe. For the dependent variable, SAHO, each country-day is coded 0 when the country has not issued a SAHO and 1 when it has. Ultimately, 41 of the 54 countries issued a SAHO during the event history. Following EHA practices [70], a country is removed from the event history starting the day after it issues a SAHO. Based on this coding scheme, Burundi, the first country to issue a SAHO (March 5), contributes 35 observations, and the 13 countries that never issued a SAHO each contribute 129 observations. As a result, the event history includes 4,138 country-day observations to be analyzed.

Because the dependent variable is dichotomous, the relationships are estimated using probit regression. Technically, then, the relationships are estimated using pooled cross-sectional time series probit, which produces maximum likelihood estimates. The probability that a country will issue a SAHO on a specified day given that it has not already issued a SAHO (i.e., its hazard rate) can be estimated from the maximum likelihood estimates. Table 1 presents the life table for these data, including the hazard rate for the pooled countries.

Besides the dependent variable, SAHO, the event history includes conventional internal determinants of policy adoption–political, social, and economic factors [39]–as well as internal determinants related to medicine that are specific to public health policy adoption [14]. The dataset also includes two external determinants of policy adoption–geographic and temporal diffusion [39]. For more details on the measures, see Box 1 and Table 2.

Medical factors are tested with a measure representing threat to health system (Hypothesis 1a), previous experience with serious infectious disease (Hypothesis 1b), and healthcare system preparation for the coronavirus pandemic (Hypothesis 1c). The first measure is threat of the disease. Given concerns about "flattening the curve," this is represented by the ratio of the cumulative number of COVID-19 cases in the country per medical doctor. This ratio ranges from 0 in the early days of the event history to almost 46 (Sao Tome and Principe) as time progressed. Because the ratio is highly positively skewed, for the regression analyses it was recoded into a series of four indicator variables with 0 cases serving as the comparison group and the remaining three groups of approximately the same size indicating progressively higher ratios. Hypothesis 1a suggests that as this ratio increases the probability that a country will issue a SAHO will also increase. This study tests robustness of this concept by replacing this ratio with cumulative COVID-19 cases per 100,000 population. This number ranges from 0 in the early

**Table 1. Life table for pooled countries.**

| Day | SAHO Issued | Risk Set | Hazard Rate |
|---|---|---|---|
| 4-Mar | 0 | 54 | 0.00 |
| 5-Mar | 1 | 54 | 0.02 |
| 15-Mar | 1 | 53 | 0.02 |
| 18-Mar | 1 | 52 | 0.02 |
| 19-Mar | 1 | 51 | 0.02 |
| 20-Mar | 2 | 50 | 0.04 |
| 21-Mar | 3 | 48 | 0.06 |
| 22-Mar | 1 | 45 | 0.02 |
| 23-Mar | 4 | 44 | 0.09 |
| 25-Mar | 2 | 40 | 0.05 |
| 26-Mar | 1 | 38 | 0.03 |
| 27-Mar | 4 | 37 | 0.11 |
| 28-Mar | 1 | 33 | 0.03 |
| 29-Mar | 1 | 32 | 0.03 |
| 30-Mar | 4 | 31 | 0.13 |
| 1-Apr | 2 | 27 | 0.07 |
| 2-Apr | 3 | 25 | 0.12 |
| 5-Apr | 1 | 22 | 0.05 |
| 6-Apr | 1 | 21 | 0.05 |
| 8-Apr | 1 | 20 | 0.05 |
| 10-Apr | 1 | 19 | 0.05 |
| 15-Apr | 1 | 18 | 0.06 |
| 18-Apr | 2 | 17 | 0.12 |
| 6-May | 1 | 15 | 0.07 |
| 8-May | 1 | 14 | 0.07 |
| Later | 0 | 13 | 0.00 |

NOTE: Day indicates day and month SAHO was issued. SAHO Issued indicates the number of SAHOs taking effect that date. Risk Set indicates the number of countries without a SAHO as of that date. Hazard Rate is the proportion of countries that had not yet issued a SAHO.

days of the event history to 417 (Djibouti) as time progressed. It, too, was recoded into a series of indicator variables due to positive skew for the regression analyses. The expectation is that as this number increases the probability that a country will issue a SAHO will also increase.

The second medical measure is experience with a serious infectious disease as represented by experience with an Ebola outbreak between 2011 and 2020. This is coded as an indicator variable such that the eight countries with Ebola outbreaks (i.e., Democratic Republic of the Congo, Guinea, Liberia, Mali, Nigeria, Senegal, Sierra Leone, and Uganda) are coded 1 and the others 0. Hypothesis 1b suggests that countries with Ebola outbreak experience will be more likely to issue a SAHO.

The third medical measure is healthcare system preparation status for the COVID-19 pandemic as represented by the WHO's Country Preparedness and Response Status report [53]. This measure is based on the self-assessed operational readiness capacity of each country and its current location on a continuum of response scenarios regarding preparedness, imported cases, and local transmission. The measure includes five categories ranging from the lowest level of country preparedness capacity, which is "no capacity" (coded 1), to the highest level, which is "sustainable" (coded 5). As shown in Table 2, the countries range from the lowest

## Box 1. Variable Descriptions

### Dependent Variable

Stay-at-home order (SAHO). Coded 1 on the effective date of a SAHO in a country and 0 on all days prior the effective date. Coded as missing on days after the effective date. A country-day is determined to have a SAHO when the government required "not leaving house with exceptions for daily exercise, grocery shopping, and 'essential' trips" (coded 2 in original data) or "not leaving house with minimal exceptions (eg allowed to leave once a week, or only one person can leave at a time, etc)" (coded 3 in original data). Source: *Oxford COVID-19 Government Response Tracker, Blavatnik School of Government [73]*.

### Independent Variables

(alternative measures for robustness checks appear in italics)

Medical Factors

- Cumulative Cases per Medical Doctor (disease risk). Daily cumulative COVID-19 cases in a country, lagged one day, divided by the number of medical doctors in that country. Because the ratio is highly positively skewed, for the regression analyses it has been recoded into a series of four indicator variables with 0 cases serving as the comparison group (n = 2542) and the remaining cases approximately evenly divided between three groups (n = 532, 533, and 531) indicating progressively higher ratios. Sources: cumulative cases as reported by the European Centre for Disease Prevention and Control (ECDC) at https://www.ecdc.europa.eu/en/publications-data/download-todays-data-geographic-distribution-covid-19-cases-worldwide; medical doctors from the WHO Global Health Observatory Data (workforce) at https://apps.who.int/gho/data/node.main.HWFGRP_0020?lang=en.

- *Cumulative Cases (disease risk). Daily cumulative COVID-19 cases in a country, lagged one day. Because the ratio is highly positively skewed, for the regression analyses it has been recoded into a series of four indicator variables with 0 cases serving as the comparison group (n = 2542) and the remaining cases approximately evenly divided between three groups (n = 534, 530, and 532) indicating progressively higher ratios. Source: cumulative cases as reported by the European Centre for Disease Prevention and Control (ECDC) at https://www.ecdc.europa.eu/en/publications-data/download-todays-data-geographic-distribution-covid-19-cases-worldwide.*

- Ebola experience. This is coded as an indicator variable such that the eight countries with Ebola outbreaks between 2011 and 2020 (i.e., Democratic Republic of the Congo, Guinea, Liberia, Mali, Nigeria, Senegal, Sierra Leone, and Uganda) are coded 1 and the others 0. Source: Centers for Disease Control and Prevention [74].

- COVID-19 Preparation Status. A categorization of countries based on their self-assessed operational readiness capacities and current location on a continuum of response scenarios related to preparedness, risk of imported cases, and transmission. Categories by increasing preparation include no capacity (coded 1), limited capacity (2), developed capacity (3), demonstrated capacity (4), and sustainable capacity (5). Source: World Health Organization [53].

Political Factors

- Regime Type. Freedom Status categorized as "not free," partly free," and "free" based on scores for political rights and civil liberties granted to citizens in 2018 by Freedom House. Source: Freedom in the World 2018: The Annual Survey of Political Rights and Civil Liberties.

- *Polity. Polity is a composite index of Autocracy/Democracy where a score of -10 indicates a strong autocracy and +10 a strong democracy. Source: Center for Systemic Peace; see* [75].

- State Fragility Index. According to Marshall and Elzinga-Marshall [76], "The State Fragility Index. . .combines scores on the eight indicators [of security, political, economic, and social effectiveness and legitimacy] and ranges from 0 'no fragility' to 25 'extreme fragility.' A country's fragility is closely associated with its state capacity to manage conflict, make and implement public policy, and deliver essential services, and its systemic resilience in maintaining system coherence, cohesion, and quality of life, responding effectively to challenges and crises, and sustaining progressive development." Source: Center for Systemic Peace.

- *Government Effectiveness. According to the World Bank Data Catalog: This measure "captures perceptions of the quality of public services, the quality of the civil service and the degree of its independence from political pressures, the quality of policy formulation and implementation, and the credibility of the government's commitment to such policies. Estimate gives the country's score on the aggregate indicator, in units of a standard normal distribution, i.e. ranging from approximately -2.5 [low effectiveness] to 2.5 [high effectiveness]." Source: World Bank variable GE.EST.*

- Colonial Legacy. Identifies the colonial history of each country as colonized by Britain (coded 0), colonized by France (coded 1), or not colonized by Britain or France (coded 2). Source: Tordoff (2016).

Social Factors

- Percent Government Healthcare Spending. Government health expenditure as a percent of total health expenditures. Mean centered in the models. Source: World Health Organization variable SH.XPD.GHED.CH.ZS.

- *Service Coverage Index. A measure of provision of selected essential health services and indication of progress toward universal health coverage. Mean centered in the models. Source: World Health Organization report on "World Health Statistics 2020: Monitoring Health for SDGs," Annex 2* [77].

- Urban Population. Urban population as percent of total population. Centered in the models. Source: United Nations Statistics variable SYB082.

- *Population Density. Population density in terms of people per km$^2$ of land area in 2018. Mean centered in the models. Source: World Bank variable EN.POP.DNST.*

Economic Factors

- GDP Per Capita. Gross Domestic Product per capita in 2017 in US dollars. Logged then centered in the models. Source: UN Stats variable SYB025.

- *Unemployment Percent. Unemployment rate as a percent of the total labor force (2019 national estimate). Mean centered in the models. Source: International Labour Organization, ILOSTAT database [78] variable SL.UEM.TOTL.NE.ZS.*

External Factors

- Geographic Diffusion. The proportion of bordering countries with a SAHO in effect by country-day. Source: calculated by the authors.

- Temporal Diffusion – Sigmoidal. An S-curve trend variable that is constructed by setting the mathematical midpoint of the event history to day number 66 then counting up to that day number from day 1 and squaring that number then counting down from that day number and squaring that number. Source: calculated by the authors.

- *Temporal Diffusion – Linear. A linear trend variable that is constructed by consecutively numbering the dates in the data set from 1 to 129. Source: calculated by the authors.*

**Table 2. Measures by country.**

| Country | SAHO Date | MEDICAL | | POLITICAL | | | | | SOCIAL | | ECON |
| | | Ebola Outbreak | Prep Status | FH Status | Polity/ (Regime) | Fragility | Govt Eff | Colonial Legacy | Govt Hlth Spend (%) | Urban (%) | GDPpc |
|---|---|---|---|---|---|---|---|---|---|---|---|
| Algeria | 23-Mar | no | 4 | NF | 2 | 11 | -0.4 | France | 66.0 | 73.2 | 4,055 |
| Angola | 27-Mar | no | 3 | NF | -2 | 11 | -1.1 | Neither | 46.3 | 66.2 | 4,274 |
| Benin | Never | no | 2 | Free | 7 | 10 | -0.6 | France | 30.0 | 47.9 | 826 |
| Botswana | 2-Apr | no | 2 | Free | 8 | 3 | 0.3 | Britain | 75.7 | 70.2 | 7,595 |
| Burkina Faso | 21-Mar | no | 2 | PF | 6 | 16 | -0.6 | France | 43.3 | 30.0 | 642 |
| Burundi | 5-Mar | no | 2 | NF | -1 | 21 | -1.4 | Neither | 24.7 | 13.4 | 290 |
| Cape Verde | 1-Apr | no | 2 | Free | 10 | 5 | 0.3 | Neither | 60.2 | 66.2 | 3,244 |
| Cameroon | Never | no | 3 | NF | -4 | 16 | -0.8 | Neither | 13.3 | 57.0 | 1,451 |
| Cent Af Rep | 8-May | no | 1 | NF | 6 | 23 | -1.7 | France | 12.8 | 41.8 | 427 |
| Chad | 2-Apr | no | 2 | NF | -2 | 19 | -1.4 | France | 15.8 | 23.3 | 719 |
| Comoros | Never | no | 1 | PF | -3 | 11 | -1.6 | France | 12.8 | 29.2 | 1,329 |
| Congo Republic | 28-Mar | no | 3 | NF | -4 | 13 | -1.2 | Neither | 40.7 | 67.4 | 2,146 |
| Cote d'Ivoire | Never | no | 3 | PF | 4 | 17 | -0.6 | France | 28.5 | 51.2 | 1,566 |
| Dem Rep Congo | 6-Apr | YES | 3 | NF | -3 | 24 | -1.6 | Neither | 10.0 | 4.6 | 462 |
| Djibouti | 23-Mar | no | 2 | NF | 3 | 12 | -0.9 | France | 47.1 | 77.9 | 1,927 |
| Egypt | 25-Mar | no | 4 | NF | -4 | 12 | -0.6 | Britain | 33.0 | 42.7 | 2,000 |
| Equatorial Guinea | 15-Mar | no | 2 | NF | -6 | 12 | -1.3 | Neither | 18.8 | 72.6 | 9,850 |
| Eritrea | 1-Apr | no | 2 | NF | -7 | 15 | -1.7 | Neither | 27.3 | 40.7 | 1,146 |
| Eswatini (Swaziland) | 27-Mar | no | 3 | NF | -9 | 8 | -0.7 | Britain | 50.8 | 24.0 | 3,224 |
| Ethiopia | Never | no | 3 | NF | 1 | 19 | -0.6 | Neither | 25.0 | 21.2 | 720 |
| Gabon | 10-Apr | no | 2 | NF | 3 | 10 | -0.8 | France | 63.3 | 89.7 | 7,220 |
| Gambia | Never | no | 2 | PF | 4 | 15 | -0.6 | Britain | 22.9 | 61.9 | 709 |
| Ghana | 30-Mar | no | 3 | Free | 8 | 11 | -0.2 | Britain | 33.5 | 56.7 | 1,056 |
| Guinea | 30-Mar | YES | 3 | PF | 4 | 18 | -1.0 | France | 17.2 | 36.5 | 802 |

*(Continued)*

**Table 2.** (Continued)

| Country | SAHO Date | MEDICAL | | POLITICAL | | | | | SOCIAL | | ECON |
| | | Ebola Outbreak | Prep Status | FH Status | Polity/ (Regime) | Fragility | Govt Eff | Colonial Legacy | Govt Hlth Spend (%) | Urban (%) | GDPpc |
|---|---|---|---|---|---|---|---|---|---|---|---|
| Guinea-Bissau | Never | no | 2 | PF | 6 | 17 | -1.5 | Neither | 8.2 | 43.8 | 723 |
| Kenya | 27-Mar | no | 3 | PF | 9 | 10 | -0.4 | Britain | 42.7 | 27.5 | 1,507 |
| Lesotho | 18-Mar | no | 2 | PF | 8 | 9 | -0.9 | Britain | 62.9 | 28.6 | 1,178 |
| Liberia | 21-Mar | YES | 3 | PF | 7 | 13 | -1.3 | Britain | 17.2 | 51.6 | 583 |
| Libya | 22-Mar | no | 2 | NF | 0 | 13 | -1.9 | Neither | 63.3 | 80.4 | 3,941 |
| Madagascar | 23-Mar | no | 2 | PF | 6 | 11 | -1.2 | France | 46.9 | 37.9 | 516 |
| Malawi | Never | no | 3 | PF | 6 | 14 | -0.7 | Britain | 30.6 | 17.2 | 340 |
| Mali | Never | YES | 3 | PF | 5 | 16 | -1.0 | France | 34.9 | 43.1 | 821 |
| Mauritania | 19-Mar | no | 2 | NF | -2 | 16 | -0.7 | France | 38.8 | 54.5 | 1,129 |
| Mauritius | 23-Mar | no | 4 | Free | 10 | 0 | 0.9 | Both | 42.9 | 40.8 | 10,564 |
| Morocco | 20-Mar | no | 3 | PF | -4 | 6 | -0.2 | France | 42.9 | 63.1 | 3,069 |
| Mozambique | Never | no | 3 | PF | 5 | 11 | -0.9 | Neither | 29.9 | 36.5 | 426 |
| Namibia | 27-Mar | no | 2 | Free | 6 | 5 | 0.1 | Neither | 46.1 | 51.0 | 5,227 |
| Niger | Never | no | 3 | PF | 5 | 18 | -0.8 | France | 33.4 | 16.5 | 378 |
| Nigeria | 29-Mar | YES | 3 | PF | 7 | 18 | -1.0 | France | 14.2 | 51.2 | 1,968 |
| Rwanda | 21-Mar | no | 3 | NF | -3 | 16 | 0.2 | Britain | 34.3 | 17.3 | 748 |
| Sao Tome & Prin | 6-May | no | 2 | Free | N/A | N/A | -0.6 | Neither | 45.6 | 73.6 | 1,921 |
| Senegal | 25-Mar | YES | 3 | Free | 7 | 10 | -0.3 | Neither | 21.0 | 47.7 | 1,332 |
| Seychelles | 8-Apr | no | 3 | PF | N/A | N/A | 0.5 | France | 73.0 | 57.1 | 15,692 |
| Sierra Leone | 5-Apr | YES | 3 | PF | 7 | 13 | 0.5 | Britain | 13.7 | 42.4 | 494 |
| Somalia | Never | no | 2 | NF | 5 | 20 | -2.2 | Britain | N/A | 45.5 | 104 |
| South Africa | 26-Mar | no | 3 | Free | 9 | 8 | 0.3 | Britain | 53.7 | 66.0 | 6,151 |
| South Sudan | 18-Apr | no | 2 | NF | 0 | 22 | -2.4 | Britain | 8.4 | 19.9 | 452 |
| Sudan | 18-Apr | no | 3 | NF | -4 | 22 | -1.6 | Britain | 18.0 | 34.9 | 2,967 |
| Tanzania | Never | no | 3 | PF | 3 | 10 | -0.8 | Britain | 43.3 | 34.5 | 933 |
| Togo | 2-Apr | no | 2 | PF | -2 | 13 | -1.1 | France | 17.7 | 42.2 | 613 |
| Tunisia | 20-Mar | no | 3 | PF | 7 | 4 | -0.1 | France | 57.1 | 69.2 | 3,474 |
| Uganda | 30-Mar | YES | 3 | PF | -1 | 16 | -0.6 | Britain | 15.7 | 24.4 | 646 |
| Zambia | 15-Apr | no | 2 | PF | 6 | 12 | -0.6 | Britain | 38.6 | 44.0 | 1,513 |
| Zimbabwe | 30-Mar | no | 3 | NF | 4 | 17 | -1.2 | Britain | 51.6 | 32.2 | 1,091 |
| MEAN [MODE] | [never] | [no] | 2.6 | [PF] | 2.6 | 13.3 | -0.8 | [Britain] | 35.2 | 45.6 | 2,374 |
| SD | | | 0.6 | | 4.9 | 5.2 | 0.7 | | 17.8 | 19.4 | 2,964 |

level of capacity in the Central African Republic and Comoros (coded 1) to the next-to-highest level in Algeria, Egypt, and Mauritius (coded 4). Hypothesis 1c suggests that countries with less capacity will be more likely to issue a SAHO.

The political measures include regime type (Hypothesis 2a), state capacity (Hypothesis 2b), and colonial legacy (Hypothesis 2c). Regime type is conceptualized in terms of limits on citizen liberties. Freedom House classifies countries as "not free," "partly free," and "free" based on their scores on indices of political and civil liberties [79]. This measure offers a direct look at the argument regarding regime practices of limiting citizen freedoms. As indicated in Table 2, Freedom House categorizes most states in Africa as not free or partly free, with a modal category of partly free. Hypothesis 2a suggests that partly free and free countries will be less likely to issue a SAHO than not free countries. This study also evaluates an alternative measure of

regime type to test the robustness of results. The Center for Systemic Peace's Polity measure scores countries in terms of autocracy versus democracy [75]. African Polity scores range from a highly authoritarian -9 in Eswatini (Swaziland) to a highly democratic 10 in Mauritius and Cape Verde. Table 2 indicates the mean on the continent is almost 3. The expectation is that authoritarian countries will be more likely to issue a SAHO.

State capacity is represented by the Center for Systemic Peace's State Fragility Index [76]. This measure assesses countries' ability to manage conflict, make policy, and deliver essential services. The scores reported in Table 2 range from 0 in Mauritius, indicating a condition of "no fragility," to a nearly maximum score of 24 in the Democratic Republic of Congo, indicating "extreme fragility." While the mean score in the region is about 13, eight of the 11 most fragile countries in the world are located in Africa. Hypothesis 2b suggests that less fragile countries will be more likely to issue SAHOs. This study tests robustness of state capacity by substituting in the model the World Bank Government Effectiveness Index. This index captures perceptions of the quality of public services and policies in countries. Scores range from -2.5 to 2.5 with higher scores indicating greater administrative capacity [80]. In Africa, the scores range from -2.4 in South Sudan to 0.9 in Mauritius with a regional mean of -0.8. The expectation is that more effective countries will be more likely to issue SAHOs.

Colonial legacy is captured by a pair of indicator variables with one variable coded 1 for countries with a French colonial history and 0 otherwise, and a second variable coded 1 for countries with neither a French nor British colonial history and 0 otherwise. The comparison group, then, is composed of the countries with a British colonial history. Table 2 shows that there are 20 African countries with a predominately British colonial history, 19 with a predominately French history, and 15 with neither. Hypothesis 2c suggests that former French colonies will be more likely to issue a SAHO than former British colonies.

The social measures include individualist versus collectivist culture (Hypothesis 3a) and population vulnerability due to social arrangements (Hypothesis 3b). Culture is captured by type of healthcare system as indicated by government expenditure on health as a percentage of total health expenditures. This reflects countries' placement of responsibility for healthcare in regard to public responsibility, indicated by higher government spending as a percentage, versus individual responsibility, indicated by lower government spending as a percentage. Currently in Africa, the average government health expenditure as a percentage of total health expenditures is about 35% as reported in Table 2, with a wide range from a minimum of about 8% in Guinea-Bissau and South Sudan to a maximum of about 76% in Botswana. Hypothesis 3a suggests that countries where government contributes more as a percentage will be more likely to issue a SAHO. This study tests robustness of culture by substituting in the WHO's Universal Health Coverage Service Coverage Index, which captures the availability in a country of essential health services without financial hardship on a scale of 0–100. In Africa, the scores range from 25 in Somalia to 78 in Algeria with a regional mean of about 48. The expectation is that countries with greater healthcare coverage will be more likely to issue SAHOs.

Population vulnerability due to social arrangements is captured by the UN's measure of percent urban population. According to UN data, the mean urban population percentage across Africa is around 45%, but that varies widely between countries. More than 70% of the populations in Algeria, Djibouti, Gabon, and Kenya live in urban areas, while less than 15% in Burundi and the Democratic Republic of Congo do the same. Hypothesis 3b suggests that the probability of a SAHO increases as the percentage of urban population increases. This study tests robustness of this concept by substituting in the UN's measure of population density per square kilometer. According to this measure, population density ranges from 3.1 people per $km^2$ in Namibia to 623.0 in Mauritius with a mean among the 54 countries of about 102. The expectation is that the probability of a SAHO increases as the population density increases.

This study further explores sociodemographic effects by testing for effects related to age, ethnic fractionalization, and education.

The effect of economic forces is assessed using the UN's measure of GDP per capita (logged). Table 2 shows that the mean GDP per capita for 2019 in Africa was about US$2,374, with Somalia on the low end at US$104 and Seychelles on the high end with US$15,692. Hypothesis 4 suggests that countries with higher GDP per capita will be less likely to issue a SAHO. This study tests robustness of this concept by substituting in the UN's measure of unemployment rate. Unemployment ranges from 0.4% in Niger to 28.2% in Lesotho with a mean among the 54 countries of 8.8%. The expectation is that the probability of a SAHO increases as the unemployment rate increases.

For completeness related to regional variation [26] and to produce more precise estimates, the analyses also attempt to account for the political, social, and economic heterogeneity of the continent by including a series of five indicator variables for region with Northern Africa serving as the comparison group for Eastern, Middle, Southern, and Western Africa. These are the regions identified by the United Nations Statistics Division for use in its publications and databases. This study also assesses regional effects at a higher level with a single indicator variable in which the Sub-Saharan countries are compared to their Northern Africa counterparts.

The last set of measures captures external factors related to geographic (Hypothesis 5a) and temporal diffusion (Hypothesis 5b). Geographic diffusion is the percentage of a country's bordering states that have previously issued a SAHO. Hypothesis 5a suggests that as this percentage increases, the probability of a SAHO will also increase. The measure of temporal diffusion captures the S-shaped social learning curve of acceleration then deceleration with the inflection point–the day on which diffusion turns from acceleration to deceleration–set to the mathematical midpoint of the event history on day 66. Hypothesis 5b suggests that as the day number gets closer to the inflection point, the probability of a SAHO will increase. This study tests the robustness of temporal diffusion using a linear learning process based on a count of days, starting with 1 on the first country-day and increasingly sequentially to 129 on the last country-day. The expectation is that the probability of a SAHO increases as the linear count increases.

## Results

As a baseline, Table 3 presents the results of 12 bivariate probit models designed to capture the zero-order correlation of each of the independent variables with the dependent variable. The models include 4,138 country-days. Model fit ranges from BICs of 402 to 497 and pseudo $R^2$s from 0.00 to 0.16. The estimates are based on robust standard errors, and all p-values are based on two-tailed tests. The results indicate that seven of the independent variables are at least marginally statistically associated with the issuance of SAHOs when the effects of the other independent variables are not controlled for. Model 1 generally suggests that greater threat as represented by more cases per medical doctor is associated with a greater probability of the issuance of a SAHO, but the effect is not monotonically positive. Model 7 suggests that as collectivism in the form of government spending on health increases so does the probability a country will issue a SAHO. Model 9 indicates that wealthier countries, not poorer ones, are more likely to issue a SAHO. Finally, Models 11 and 12 indicate that SAHOs diffused geographically (p = 0.08) and temporally. The remaining models indicate that the bivariate relationships are not statistically meaningful.

Table 4 presents the results of nine multiple variate probit regression models. Model 1 is offered as the theoretically best model, which includes the variables that most effectively capture the factors expected to affect the issuance of a SAHO in an African country. Models A1 to

**Table 3. Bivariate probit models with average marginal effects.**

| | Hyp | (1) | (2) | (3) | (4) | (5) | (6) | (7) | (8) | (9) | (10) | (11) | (12) |
|---|---|---|---|---|---|---|---|---|---|---|---|---|---|
| ***MEDICAL*** | | | | | | | | | | | | | |
| CasesMD.1 | 1a | 1.126*** | | | | | | | | | | | |
| | | (0.186) | | | | | | | | | | | |
| | | [0.032***] | | | | | | | | | | | |
| CasesMD.2 | 1a | 1.099*** | | | | | | | | | | | |
| | | (0.187) | | | | | | | | | | | |
| | | [0.030***] | | | | | | | | | | | |
| CasesMD.3 | 1a | 0.281 | | | | | | | | | | | |
| | | (0.283) | | | | | | | | | | | |
| | | [0.002] | | | | | | | | | | | |
| Ebola | 1b | | 0.114 | | | | | | | | | | |
| | | | (0.159) | | | | | | | | | | |
| | | | [0.003] | | | | | | | | | | |
| Prep Status | 1c | | | 0.037 | | | | | | | | | |
| | | | | (0.090) | | | | | | | | | |
| | | | | [0.001] | | | | | | | | | |
| ***POLITICAL*** | | | | | | | | | | | | | |
| Regime: Part Free | 2a | | | | -0.208 | | | | | | | | |
| | | | | | (0.130) | | | | | | | | |
| | | | | | [-0.005] | | | | | | | | |
| Regime: Free | 2a | | | | 0.008 | | | | | | | | |
| | | | | | (0.162) | | | | | | | | |
| | | | | | [0.000] | | | | | | | | |
| Fragility | 2b | | | | | -0.015 | | | | | | | |
| | | | | | | (0.012) | | | | | | | |
| | | | | | | [-0.000] | | | | | | | |
| Colonial: France | 2c | | | | | | -0.063 | | | | | | |
| | | | | | | | (0.137) | | | | | | |
| | | | | | | | [-0.002] | | | | | | |
| Colonial: Neither | 2c | | | | | | -0.051 | | | | | | |
| | | | | | | | (0.145) | | | | | | |
| | | | | | | | [-0.001] | | | | | | |
| ***SOCIAL*** | | | | | | | | | | | | | |
| Govt Hlth Spdg % | 3a | | | | | | | 0.007* | | | | | |
| | | | | | | | | (0.003) | | | | | |
| | | | | | | | | [0.000*] | | | | | |
| Urban Pop % | 3b | | | | | | | | 0.004 | | | | |
| | | | | | | | | | (0.003) | | | | |
| | | | | | | | | | [0.000] | | | | |
| ***ECONOMIC*** | | | | | | | | | | | | | |
| GDP per capita | 4 | | | | | | | | | 0.143** | | | |
| | | | | | | | | | | (0.054) | | | |
| | | | | | | | | | | [0.004*] | | | |
| ***REGION*** | | | | | | | | | | | | | |
| East | | | | | | | | | | | -0.301 | | |
| | | | | | | | | | | | (0.195) | | |
| | | | | | | | | | | | [-0.010] | | |

*(Continued)*

**Table 3.** (Continued)

| | Hyp | (1) | (2) | (3) | (4) | (5) | (6) | (7) | (8) | (9) | (10) | (11) | (12) |
|---|---|---|---|---|---|---|---|---|---|---|---|---|---|
| Middle | | | | | | | | | | | -0.164 | | |
| | | | | | | | | | | | (0.212) | | |
| | | | | | | | | | | | [-0.006] | | |
| South | | | | | | | | | | | 0.002 | | |
| | | | | | | | | | | | (0.244) | | |
| | | | | | | | | | | | [0.000] | | |
| West | | | | | | | | | | | -0.333+ | | |
| | | | | | | | | | | | (0.200) | | |
| | | | | | | | | | | | [-0.010] | | |
| *EXTERNAL* | | | | | | | | | | | | | |
| Diffusion: Geographic | 5a | | | | | | | | | | | 0.003+ | |
| | | | | | | | | | | | | (0.001) | |
| | | | | | | | | | | | | [0.000*] | |
| Diffusion: Temporal, S | 5b | | | | | | | | | | | | 0.000*** |
| | | | | | | | | | | | | | (0.000) |
| | | | | | | | | | | | | | [0.000***] |
| Constant | | -2.953*** | -2.347*** | -2.329*** | -2.244*** | -2.329*** | -2.295*** | -2.329*** | -2.330*** | -2.331*** | -2.104*** | -2.384*** | -3.273*** |
| | | (0.154) | (0.064) | (0.058) | (0.088) | (0.059) | (0.091) | (0.059) | (0.059) | (0.059) | (0.164) | (0.067) | (0.131) |
| *N* | | 4138 | 4138 | 4138 | 4138 | 4138 | 4138 | 4138 | 4138 | 4138 | 4138 | 4138 | 4138 |
| $X^2$ | | 48.6*** | 0.5 | 0.2 | 3.0 | 1.7 | 0.2 | 4.1* | 1.2 | 7.1* | 5.1 | 3.2+ | 98.1*** |
| *BIC* | | 427.1 | 476.1 | 476.5 | 481.8 | 474.7 | 484.7 | 472.4 | 475.2 | 470.3 | 496.8 | 474.6 | 401.6 |
| pseudo $R^2$ | | 0.14 | 0.00 | 0.00 | 0.01 | 0.00 | 0.00 | 0.01 | 0.00 | 0.01 | 0.01 | 0.00 | 0.16 |

NOTES: Robust standard errors in parentheses. Average marginal effects in square brackets. + p < .10

* p < .05

** p < .01

*** p < .001 (2-tailed tests).

NOTE: DV is SAHO where 1 = SAHO issued, 0 = not issued. BIC indicates Bayesian Information Criterion.

A8 provide robustness checks for several of the hypothesized relationships by respecifying Model 1 using alternative measures. The models include 4,138 country-days. Each model is statistically significant. Model fit ranges from BICs of 485 to 529 and pseudo $R^2$s from 0.21 to 0.26. The estimates are based on robust standard errors due to evidence of heteroskedasticity, and all p-values are based on two-tailed tests. Tests suggest that collinearity is not unduly affecting the regression coefficients or standard errors.

Fig 2 presents an overview of the results in terms of the average marginal effects (AMEs) of each measure derived from Model 1. This reporting focuses on AMEs because probit estimates provide limited substantive information other than direction and statistical significance, although all probit estimates are reported in Table 4. AMEs indicate the mean marginal effect of each variable for all observations. In this case the AMEs show the change in probability of a country issuing a SAHO on any given day assuming it has not already issued a SAHO based on a one-unit change in each continuous measure and a one-unit change from the comparison group for each indicator variable. The figure and results reported in Table 4 indicate that six of the 19 individual measures evaluated in this study yield a statistically significant relationship at conventional levels with countries' decisions to issue SAHOs (note that temporal diffusion is

**Table 4. Multivariate probit models with average marginal effects.**

| | | (1) | (2) | (3) | (4) | (5) | (6) | (7) | (8) | (9) |
|---|---|---|---|---|---|---|---|---|---|---|
| | Hyp | Model 1 | A1 | A2 | A3 | A4 | A5 | A6 | A7 | A8 |
| *MEDICAL* | | | | | | | | | | |
| CasesMD.1 | 1a | 0.855*** | | 0.861*** | 0.788*** | 0.842*** | 0.919*** | 0.891*** | 0.848*** | 1.283*** |
| | | (0.221) | | (0.225) | (0.223) | (0.216) | (0.234) | (0.227) | (0.217) | (0.242) |
| | | [0.014**] | | [0.015**] | [0.013**] | [0.014**] | [0.015**] | [0.015**] | [0.015**] | [0.032***] |
| CasesMD.2 | 1a | 0.933*** | | 0.957*** | 0.928*** | 0.958*** | 1.006*** | 0.937*** | 0.918*** | 1.353*** |
| | | (0.233) | | (0.243) | (0.243) | (0.238) | (0.234) | (0.239) | (0.223) | (0.310) |
| | | [0.017**] | | [0.018**] | [0.018**] | [0.018**] | [0.018**] | [0.017**] | [0.017**] | [0.037*] |
| CasesMD.3 | 1a | 0.942** | | 0.857* | 1.011** | 0.943** | 0.959** | 0.927* | 0.808* | 0.501 |
| | | (0.355) | | (0.366) | (0.377) | (0.343) | (0.363) | (0.372) | (0.364) | (0.459) |
| | | [0.018] | | [0.014] | [0.022] | [0.018] | [0.017] | [0.017] | [0.013] | [0.004] |
| Ebola | 1b | 0.895** | 0.894** | 0.677* | 0.786** | 0.854** | 0.924** | 0.908** | 0.917** | 0.996** |
| | | (0.291) | (0.286) | (0.264) | (0.287) | (0.298) | (0.299) | (0.286) | (0.282) | (0.327) |
| | | [0.019**] | [0.019**] | [0.014*] | [0.017**] | [0.018**] | [0.019**] | [0.019**] | [0.019**] | [0.022**] |
| Prep Status | 1c | -0.408* | -0.390* | -0.309* | -0.379+ | -0.364* | -0.360** | -0.427* | -0.398* | -0.492** |
| | | (0.170) | (0.167) | (0.158) | (0.198) | (0.160) | (0.132) | (0.180) | (0.160) | (0.159) |
| | | [-0.009*] | [-0.008*] | [-0.007+] | [-0.008+] | [-0.008*] | [-0.007*] | [-0.009*] | [-0.008*] | [-0.011**] |
| *POLITICAL* | | | | | | | | | | |
| Regime: Part Free | 2a | -0.762** | -0.757** | | -0.524* | -0.777** | -0.735** | -0.795** | -0.659** | -0.735** |
| | | (0.267) | (0.270) | | (0.257) | (0.268) | (0.275) | (0.289) | (0.216) | (0.268) |
| | | [-0.022*] | [-0.021*] | | [-0.013+] | [-0.023*] | [-0.021*] | [-0.024+] | [-0.017*] | [-0.022+] |
| Regime: Free | 2a | -0.845* | -0.837* | | -0.527 | -0.911* | -0.832* | -0.935* | -0.644+ | -0.728* |
| | | (0.385) | (0.390) | | (0.368) | (0.376) | (0.380) | (0.424) | (0.333) | (0.369) |
| | | [-0.023+] | [-0.022+] | | [-0.013] | [-0.025*] | [-0.022+] | [-0.025*] | [-0.017+] | [-0.021*] |
| Fragility | 2b | -0.067+ | -0.062+ | -0.009 | | -0.083* | -0.040 | -0.074* | -0.065* | -0.061+ |
| | | (0.034) | (0.035) | (0.025) | | (0.039) | (0.037) | (0.037) | (0.033) | (0.035) |
| | | [-0.001+] | [-0.001+] | [-0.000] | | [-0.002*] | [-0.001] | [-0.002+] | [-0.001+] | [-0.001+] |
| Colonial: France | 2c | -0.134 | -0.141 | -0.106 | -0.069 | -0.106 | -0.093 | -0.166 | -0.166 | -0.379+ |
| | | (0.213) | (0.213) | (0.206) | (0.206) | (0.218) | (0.198) | (0.210) | (0.190) | (0.217) |
| | | [-0.003] | [-0.003] | [-0.002] | [-0.001] | [-0.002] | [-0.002] | [-0.003] | [-0.003] | [-0.008+] |
| Colonial: Neither | 2c | 0.006 | -0.004 | 0.072 | 0.060 | 0.033 | 0.041 | 0.017 | -0.064 | -0.099 |
| | | (0.199) | (0.198) | (0.190) | (0.197) | (0.198) | (0.193) | (0.207) | (0.189) | (0.191) |
| | | [0.000] | [-0.000] | [0.002] | [0.001] | [0.001] | [0.001] | [0.000] | [-0.001] | [-0.003] |
| *SOCIAL* | | | | | | | | | | |
| Govt Hlth Spdg % | 3a | 0.006 | 0.006 | 0.012 | 0.009 | | 0.010 | 0.008 | 0.008 | 0.012 |
| | | (0.009) | (0.009) | (0.010) | (0.009) | | (0.011) | (0.008) | (0.009) | (0.009) |
| | | [0.000] | [0.000] | [0.000] | [0.000] | | [0.000] | [0.000] | [0.000] | [0.000] |
| Urban Pop % | 3b | -0.007 | -0.007 | -0.001 | -0.002 | -0.005 | | -0.005 | -0.008 | -0.008 |
| | | (0.007) | (0.007) | (0.007) | (0.006) | (0.006) | | (0.007) | (0.006) | (0.007) |
| | | [-0.000] | [-0.000] | [-0.000] | [-0.000] | [-0.000] | | [-0.000] | [-0.000] | [-0.000] |
| *ECONOMIC* | | | | | | | | | | |
| GDP per capita | 4 | 0.043 | 0.015 | 0.007 | 0.050 | 0.105 | -0.040 | | 0.034 | 0.056 |
| | | (0.128) | (0.134) | (0.149) | (0.143) | (0.116) | (0.152) | | (0.125) | (0.127) |
| | | [0.001] | [0.000] | [0.000] | [0.001] | [0.002] | [-0.001] | | [0.001] | [0.001] |
| *REGION* | | | | | | | | | | |
| East | | -0.556+ | -0.572+ | -0.527 | -0.501 | -0.595 | -0.652+ | -0.689* | | -0.517 |
| | | (0.330) | (0.333) | (0.329) | (0.353) | (0.390) | (0.333) | (0.328) | | (0.327) |

(*Continued*)

**Table 4.** (Continued)

| | Hyp | (1) Model 1 | (2) A1 | (3) A2 | (4) A3 | (5) A4 | (6) A5 | (7) A6 | (8) A7 | (9) A8 |
|---|---|---|---|---|---|---|---|---|---|---|
| | | [-0.017] | [-0.017] | [-0.016] | [-0.016] | [-0.019] | [-0.020] | [-0.021] | | [-0.015] |
| Middle | | -0.729* | -0.752* | -0.532 | -0.755+ | -0.842* | -0.774* | -0.784* | | -0.430 |
| | | (0.335) | (0.336) | (0.332) | (0.392) | (0.376) | (0.351) | (0.337) | | (0.312) |
| | | [-0.020] | [-0.020] | [-0.016] | [-0.020] | [-0.023] | [-0.022] | [-0.023+] | | [-0.013] |
| South | | -0.365 | -0.357 | -0.208 | -0.207 | -0.329 | -0.080 | -0.137 | | -0.462 |
| | | (0.422) | (0.421) | (0.367) | (0.407) | (0.435) | (0.339) | (0.468) | | (0.411) |
| | | [-0.013] | [-0.013] | [-0.008] | [-0.008] | [-0.013] | [-0.004] | [-0.006] | | [-0.014] |
| West | | -0.397 | -0.411 | -0.574+ | -0.531 | -0.485 | -0.505 | -0.503 | | -0.308 |
| | | (0.365) | (0.382) | (0.343) | (0.403) | (0.417) | (0.390) | (0.356) | | (0.360) |
| | | [-0.013] | [-0.014] | [-0.017] | [-0.017] | [-0.017] | [-0.017] | [-0.018] | | [-0.010] |
| *EXTERNAL* | | | | | | | | | | |
| Diffusion: Geographic | 5a | -0.004 | -0.005 | -0.005 | -0.006+ | -0.005 | -0.004 | -0.004 | -0.004 | -0.002 |
| | | (0.004) | (0.003) | (0.004) | (0.003) | (0.004) | (0.003) | (0.004) | (0.003) | (0.004) |
| | | [-0.000] | [-0.000] | [-0.000] | [-0.000+] | [-0.000] | [-0.000] | [-0.000] | [-0.000] | [-0.000] |
| Diffusion: Temporal, S | 5b | 0.000*** | 0.000*** | 0.000*** | 0.000*** | 0.000*** | 0.000*** | 0.000*** | 0.000*** | |
| | | (0.000) | (0.000) | (0.000) | (0.000) | (0.000) | (0.000) | (0.000) | (0.000) | |
| | | [0.000***] | [0.000***] | [0.000***] | [0.000***] | [0.000***] | [0.000***] | [0.000***] | [0.000***] | |
| *ROBUSTNESS* | | | | | | | | | | |
| Cumulative Cases.1 | 1a | | 0.775** | | | | | | | |
| | | | (0.236) | | | | | | | |
| | | | [0.012**] | | | | | | | |
| Cumulative Cases.2 | 1a | | 0.919*** | | | | | | | |
| | | | (0.228) | | | | | | | |
| | | | [0.017**] | | | | | | | |
| Cumulative Cases.3 | 1a | | 1.027*** | | | | | | | |
| | | | (0.287) | | | | | | | |
| | | | [0.022*] | | | | | | | |
| Polity | 2a | | | -0.024 | | | | | | |
| | | | | (0.023) | | | | | | |
| | | | | [-0.001] | | | | | | |
| Govt Effectiveness | 2b | | | | 0.208 | | | | | |
| | | | | | (0.225) | | | | | |
| | | | | | [0.004] | | | | | |
| Service Coverage | 3a | | | | | -0.007 | | | | |
| | | | | | | (0.017) | | | | |
| | | | | | | [-0.000] | | | | |
| Population Density | 3b | | | | | | 0.002 | | | |
| | | | | | | | (0.001) | | | |
| | | | | | | | [0.000] | | | |
| Unemployment % | 4 | | | | | | | -0.019 | | |
| | | | | | | | | (0.020) | | |
| | | | | | | | | [-0.000] | | |
| Sub-Sahara | | | | | | | | | -0.571* | |
| | | | | | | | | | (0.291) | |
| | | | | | | | | | [-0.012+] | |
| Diffusion: Temp, Linear | 5a | | | | | | | | | 0.006 |

(*Continued*)

**Table 4.** (Continued)

|  | Hyp | (1)<br>Model 1 | (2)<br>A1 | (3)<br>A2 | (4)<br>A3 | (5)<br>A4 | (6)<br>A5 | (7)<br>A6 | (8)<br>A7 | (9)<br>A8 |
|---|---|---|---|---|---|---|---|---|---|---|
|  |  |  |  |  |  |  |  |  |  | (0.005) |
|  |  |  |  |  |  |  |  |  |  | [0.000] |
| Constant |  | -2.909*** | -2.896*** | -3.238*** | -2.859*** | -2.846*** | -3.011*** | -2.836*** | -2.852*** | -2.578*** |
|  |  | (0.325) | (0.332) | (0.309) | (0.470) | (0.357) | (0.301) | (0.345) | (0.301) | (0.292) |
| $N$ |  | 4138 | 4138 | 4138 | 4138 | 4138 | 4138 | 4138 | 4138 | 4138 |
| $X^2$ |  | 100.3*** | 114.3*** | 91.3*** | 115.9*** | 100.2*** | 110.2*** | 99.8** | 107.3*** | 96.7*** |
| $BIC$ |  | 508.1 | 507.3 | 506.1 | 511.1 | 508.6 | 505.1 | 507.2 | 484.7 | 528.6 |
| Pseudo $R^2$ |  | 0.26 | 0.26 | 0.24 | 0.25 | 0.26 | 0.26 | 0.26 | 0.25 | 0.21 |

NOTES: Robust standard errors in parentheses. Average marginal effects in square brackets. + $p < .10$

* $p < .05$

** $p < .01$

*** $p < .001$ (2-tailed tests).

NOTE: DV is SAHO where 1 = SAHO issued, 0 = not issued. BIC indicates Bayesian Information Criterion.

statistically significant, but the change in probability is small and not discernible on this figure), and two yield marginally significant relationships ($p < 0.10$).

The models include three medical measures related to Hypotheses 1a to 1c. Offering considerable support for H1a, the results suggest that increased threat of disease was associated with increased probability of issuing a SAHO, which is consistent with efforts to flatten their pandemic curves to avoid threat to their healthcare systems. This is represented by a series of

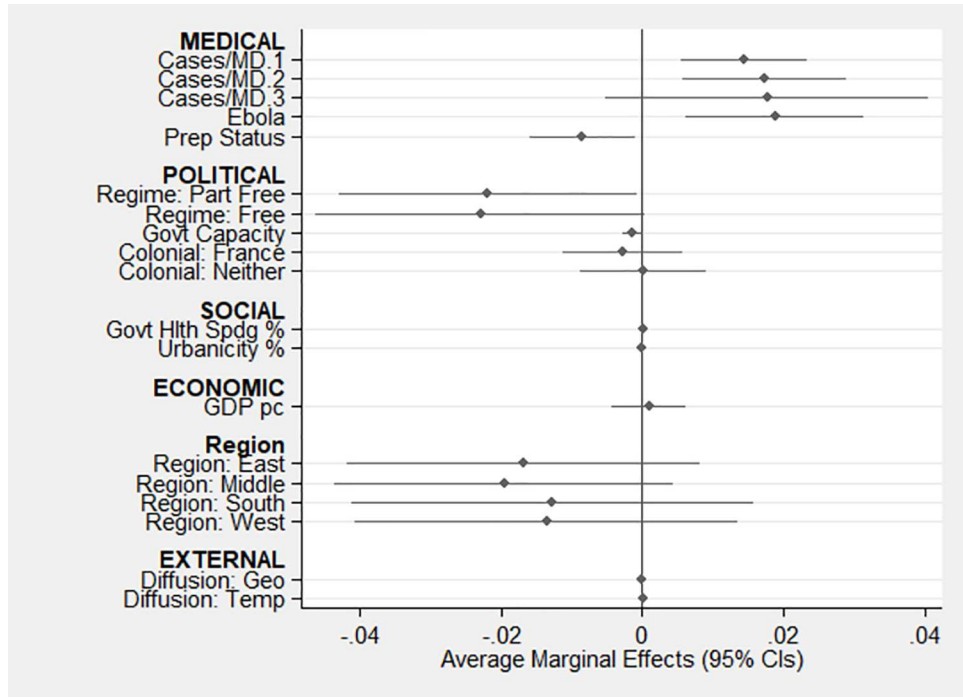

**Fig 2. Average marginal effects derived from Model 1.** NOTE: Temporal diffusion is statistically significant ($p < 0.001$), but the effect is not discernible in this figure.

indicator variables that represent the ratio of COVID-19 cases to medical doctors by country and day with the comparison group being 0 cases per medical doctor. The AMEs derived from Model 1 indicate that countries in the lowest group of cases per doctor ($>$ 0.0000–0.0068) were 1.4 percentage points more likely to issue a SAHO than when there are no cases per doctor and 1.7 percentage points more likely when in the middle group (0.0069–0.1080). The highest group (0.1081–15.8182) was 1.8 percentage points more likely, but the large standard error reduced the effect to statistical insignificance. Further analyses indicate there is no statistical difference between any of the three groups with cases. Table 4 shows that all eight of the pertinent models yielded a statistically significant effect in this regard as did Model A1, which substituted cases per 100,000 population for cases per MD. Further, previous experience with deadly infectious disease, the second medical factor, also likely played a role as suggested by H1b. The eight countries with substantial Ebola cases since 2011 were statistically more likely to issue a SAHO in all eight models. In particular, Model 1 suggests these countries were 1.9 percentage points more likely than the other countries. Finally, preparation status also played a role as suggested by H1c. Model 1 implies that less prepared countries were more likely to issue a SAHO, and the effect is almost one percentage point for each level of drop in preparedness. Overall, given the support for H1a, H1b, and H1c, the results indicate that medical considerations played nontrivial roles in countries' decisions to issue SAHOs in response to COVID-19.

The models include three political factors, which are related to Hypotheses 2a to 2c. In terms of Hypothesis 2a, the regression coefficients estimated in Model 1 suggest that Freedom House "partly free" and "free" regimes were statistically less likely to issue a SAHO than "not free" regimes (comparison group). In terms of AMEs, partly free regimes were 2.2 percentage points less likely to issue a SAHO than not free regimes, and free regimes were 2.3 percentage points less likely. This relationship is robust across all but one of the remaining pertinent models (A1 and A3-A8). Additional analyses using Wald tests indicate there is no statistical difference between the "partly free" and "free" regimes in any of the models (e.g., Model 1: $\chi^2(1) =$ 0.08, p = 0.78). Model A2, which replaces Freedom House category with Polity score to test the robustness of the effect, fails to confirm the statistically significant relationship, although the signs on the coefficient and AME are in the expected negative direction. Overall, while Model 1 supports Hypothesis 2a as do the other models using the Freedom House measure, Model A1 with the alternative measure of regime does not. Together, this suggests somewhat mixed support for Hypothesis 2a.

The second political factor, state capacity, appears to have played an unclear role in the countries' decisions to issue SAHOs. Consistent with H2b, Model 1 indicates that as state fragility increased, a country's likelihood of issuing a SAHO decreased, but the effect was marginally statistically significant (p $<$ 0.06). Six of the eight pertinent models yield a similar relationship. On the other hand, the Government Effectiveness Index in Model A3 is statistically unrelated to the dependent variable. Further, and contrary to H2c, there is no evidence that colonial legacies played a role, including comparing countries with French colonial legacies to countries with no British or French colonial legacy. Overall, given the mixed support for H2a, limited support for H2b, and lack of support for H2c, it appears that political forces played a limited role at most in the countries' decisions to issue SAHOs.

The models also account for social factors related to the perception of health as a public/collective versus private/individual responsibility and living arrangements. Contrary to H3a, countries in which healthcare is provided in a more collective manner, through a greater government contribution to healthcare expenditures, were not statistically more likely to issue a SAHO according to any of the models, including Model A4, which replaced government healthcare expenditures with Service Coverage Index score. Moreover, and contrary to H3b,

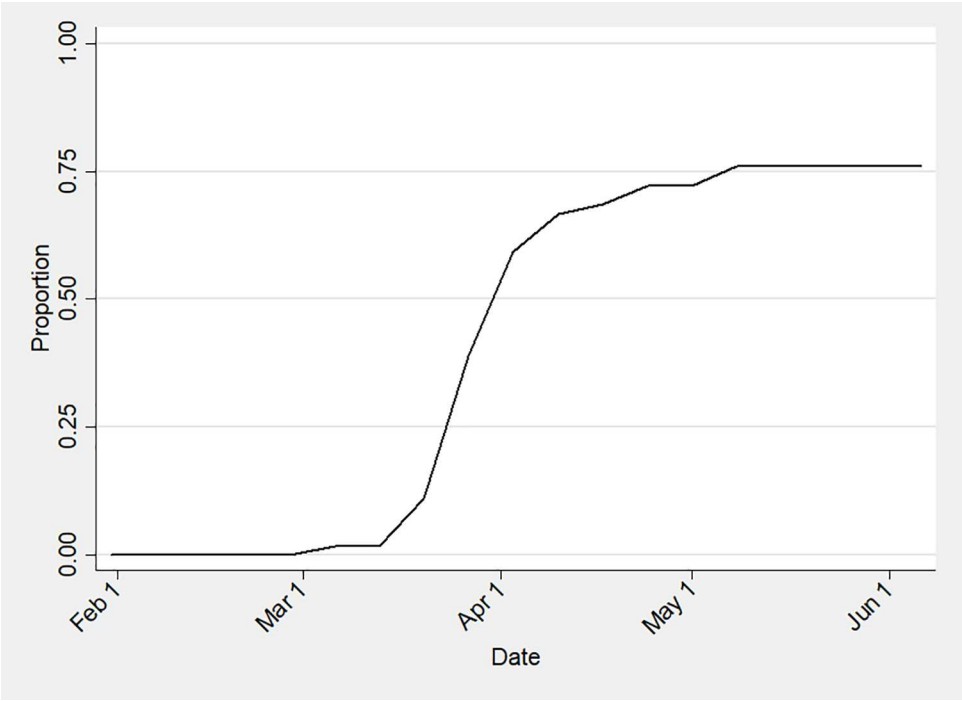

**Fig 3. Cumulative distribution of SAHOs by date.**

the same can be said for the models addressing living conditions in terms of greater urban population and its alternative, population density. Although not detailed here, additional analyses of other sociodemographic measures also failed to yield statistically significant effects including measures of ethnic fractionalization [81], religious fractionalization [81], net migration, years of compulsory education, and percent population 65 years old and older. Overall, the results suggest that social factors played no role in the countries' decisions to issue SAHOs.

Contrary to H4, GDP per capita failed to achieve statistical significance in any of the models in which it was specified as was the case for unemployment rate, which served as an alternative specification in Model A6. Overall, the results suggest that economic considerations played no role in the countries' decisions to issue SAHOs.

Finally, the models also capture factors external to the countries that may have played a role, in particular geographic and temporal diffusion. Contrary to H5a, only one of the models suggests that the percentage of bordering countries previously issuing a SAHO had a statistically discernible effect on the probability a country would issue its own SAHO. On the other hand, the results are consistent with H5b. Fig 3 presents the cumulative distribution of issued SAHOs over time and confirms the expected S-shaped curve. Further, all the models indicate SAHOs diffused temporally following an S-shaped curve, though Model A7 suggests that diffusion did not fit a linear pattern. In some way, though, the failed robustness check suggests that it was not the simple passage of time that played a role, it was the distinct social learning curve that accounted for the effect. Overall, these analyses indicate that a specific pattern of temporal diffusion played a significant role, but geographic diffusion did not.

Further, Model A8 indicates that Sub-Saharan countries were 1.2 percentage points less likely to issue a SAHO than their Northern Africa counterparts, but the results in Model 1 and the remaining models suggest the effect is primarily driven by countries in Eastern and Middle Africa.

### Relative effects of the five broad factors

It is also useful to assess the relative effects of the five broad factors in public health policymaking to gain an understanding of which considerations leaders prioritized in their decision-making regarding coronavirus SAHOs. Table 5 reports a number of analyses that offer some guidance on the relative effects of the medical, political, social, economic, and external factors. The first analyses present postestimation tests of Model 1 of the joint significance of the variables that compose each factor. These Wald tests suggest that the variables that compose the medical factor as a whole are statistically significant as are the variables that compose the external factor, while the variables composing the political, social, and economic factors are not.

The remaining analyses in Table 5 are based on models that include only the variables composing each factor. For instance, the analysis of the political factor includes a model using only the regime type, fragility, and colonial legacy measures. The first of these analyses reports Wald $\chi^2$ tests, which indicate, again, that the medical and external variables are statistically significant but also that the economic variable is statistically significant (as in Table 3). While being a less-than-ideal measure of model fit, the next column reports McFadden pseudo $R^2$ for the models using each group of variables. Again, the models using the medical and external factors stand out in their explanatory power from the other factors. The final analyses use the Bayesian Information Criterion (BIC), which scores models based on the uncertainty of their estimates [82]. Estimate uncertainty decreases as BIC values decrease, therefore, models with lower BIC values are more desirable. The table shows that the best model by far is generated by the group of external measures. The BIC for this model is almost 100 points less/better than the worst model (political) and more than 35 points less/better than the next best model (medical). To put this in context, Raftery [82] states that BIC score differences of 0 to 2 points represent "weak" evidence for preferring one model over another, differences of 2 to 6 points represent "positive" evidence, differences of 6 to 10 points represent "strong" evidence, and differences of 10 points or more represent "very strong" evidence. By this analysis, then, the external factor played the largest relative role by far in the decision to issue SAHOs followed by the medical factor, which was followed at a large distance by the economic then social and political factors.

### Conclusion

The objective of this study is to identify factors related to African countries' issuance of SAHOs in response to the COVID-19 pandemic. The analyses capture conventional measures of public health policy adoption related to medical, political, social, economic, and external factors. To the authors' knowledge, this is the first study to comprehensively assess how the broad range of public health policymaking factors affected the issuance of SAHOs in Africa. More broadly, it sheds light on how leaders in Africa prioritized competing factors in their

**Table 5. Relative effects of the broad factors.**

| Factors | Joint Significance | $X^2$ | Pseudo R$^2$ | BIC |
|---|---|---|---|---|
| Medical | 25.81* | 64.74* | 0.15 | 442.45 |
| Political | 8.60 | 8.40 | 0.02 | 502.53 |
| Social | 1.05 | 4.17 | 0.01 | 480.67 |
| Economic | 0.11 | 7.08* | 0.01 | 470.27 |
| External | 29.32* | 80.86* | 0.17 | 405.97 |

* p < 0.05 + p < 0.10 two tailed.

**Table 6. Summary of results by hypothesis.**

| Hypothesis | Measure | Model 1 Support | Alt Model Support |
|---|---|---|---|
| MEDICAL | | | |
| 1a | Threat increase | yes | yes |
| 1b | Ebola | yes | NA |
| 1c | Preparation status | yes | NA |
| POLITICAL | | | |
| 2a | Regime type | yes | no |
| 2b | Governance capacity | yes | no |
| 2c | Colonial legacy | no | NA |
| SOCIAL | | | |
| 3a | Collectivism | no | no |
| 3b | Population risk | no | no |
| ECONOMIC | | | |
| 4 | Economy | no | no |
| EXTERNAL | | | |
| 5a | Geographic diffusion | no | NA |
| 5b | Temporal diffusion | yes | no |

response to the coronavirus pandemic. Table 6 summarizes the results by hypothesis. It suggests that medical considerations played consistent roles, while external and political considerations played inconsistent roles. It also suggests that social and economic considerations played little to no role.

The findings in Table 4 consistently indicate that in terms of immediate medical conditions, increased disease threat–the strain imposed on the healthcare system as indicated by the number of cases per medical doctor and, as a robustness check, the number of cases per capita–was associated with a greater probability that a country would be locked down in support of H1a. In particular, the effect of cases per doctor is consistent with public health experts' calls to "flatten the curve" to reduce the probability that healthcare systems would be overwhelmed by patients [48]. They also show that experience with deadly infectious disease was consistently related to countries' responses as asserted by H1b. Countries with publicly reported Ebola cases and outbreaks since 2011 were more likely to issue a SAHO than countries without, which supports the assertion that countries learn from their experiences (e.g., [49]). Further, there is consistent statistical evidence that as a country's pandemic preparation status increased, the probability it would issue a SAHO decreased as suggested by H1c, which is consistent with the proposition that more prepared countries had other viable response options than issuing a SAHO (e.g., [51]). These results and those reported in Table 5, which showed robust combined effects for the medical variables across several tests, suggest that medical considerations played a substantial role in decisions to issue SAHOs.

The results presented in Table 4 also suggest that political factors played something of a role in SAHO decisions. The evidence from several models shows that more authoritarian regimes were more likely to issue SAHOs than more democratic regimes as proposed by H2a, but the robustness check did not confirm the effect. It seems concentration of power may have made it easier to issue disruptive policies like SAHOs (e.g., [54]), but the effect is still open to further investigation. There is also evidence in some of the models that state administrative capacity may have mattered as countries with greater capacity were more likely to issue a SAHO as suggested by H2b, but, again, the effect was not confirmed by the robustness check. Concerns about undermining state legitimacy [57], then, remain open for more research. There is no

evidence to support H2c that colonial legacy played a role, as most of the related AMEs were very close to 0 and statistically insignificant. This may be consistent with recent findings that the influence of colonial institutions in Africa is waning over time [83]; but see [58]). In all, these results hint at a role for political considerations but, combined with the results in Table 5, which showed little to no joint effect for the political variables, suggest that the hypothesized political effects did not play a significant role in the decision to issue SAHOs despite arguments about the primacy of political considerations in public health policy [12, 13].

Further, there is no evidence in Table 4 that social factors mattered, whether in the form of cultural orientation toward shared responsibility for public health (H3a) or population vulnerability (H3b). Additional analyses using other socio-demographic measures also failed to detect a meaningful effect. Contrary to H4, economic considerations appear to have played no role, despite the bivariate results presented in Table 3 and previous research suggesting economic conditions play a key role in public health policy due to the extraordinary costs associated with healthcare (e.g., [36, 63]). Finally, the external factor, particularly temporal diffusion, was associated with substantial effects as suggested by H5b. Specifically, the results suggest the probability of a SAHO increased following a common S-curve pattern often associated with a social learning process (e.g., [45]). While the effect was not robust to the alternative linear specification of temporal diffusion proposed in H5a, it may be that the lack of confirmation underscores the importance of the social learning curve and not the simple passage of time. Significantly, many of the analyses, particularly in Table 5, imply that this form of temporal diffusion was a major factor, if not *the* major factor, in decisions to issue SAHOs.

Overall, the results, particularly from Table 5 regarding relative effects, suggest that medical considerations were a priority for decision makers, but that temporal diffusion, an external factor, played the most significant role. On the other hand, the evidence is mixed at best regarding the effect of political considerations and clear that social and economic factors played little to no role. The lack of effects for some factors, particularly the social and economic factors, may be related to the speed with which the SAHOs were issued. The 41 countries that locked down issued their SAHOs within a 64-day period. The quickly mounting worldwide cases and deaths drew significant public attention, which may have pushed the pandemic into the realm of Christensen et al.'s [46] "wicked" crises composed of transboundary, unique, and uncertain threat, which made an accelerated response rational [45]. Over such an abbreviated and anxiety provoking period of time, it is likely that longer-term factors, such as social and economic issues, would be uncertain and incompletely understood relative to the medical considerations and, therefore, devalued relative to the medical considerations.

When evaluating this study's findings, one must also account for its limitations. Among them, the full event history record only covers about five months, which is an extremely short period of time in policy adoption research. The urgent nature of the issue drove this timeframe [45], though, and there is no reason to believe that the EHA methodology is unable to accommodate such a brief period [70]. Further, while the results address 54 nations and effects on more than 1 billion people and account for multiple factors, the nations are quite heterogenous and such a broad perspective may mask more important effects that are unique to specific nations or regions. The results do not account for the effects of subnational policies that may impact national policy [84], nor do the data capture the effects of non-public data such as leaders' personal incentives and fears or even the limitations on the information they had to make their decisions in the early phase of the COVID-19 pandemic. Moreover, much of the data come from authoritarian regimes and/or governments with very limited state capacity. As such, the validity and integrity of the data should be rigorously considered [85]. Finally, previous research indicates that the media can have an important effect on policymaking (e.g.,

[86]), but resource limitations, financial as well as temporal, prevented the collection of media data.

In conclusion, this study gives researchers and policy makers an early view of a wide range of government responses to the COVID-19 pandemic. In particular, it gives a view of the policy responses from a region of the world facing a disproportionate share of the global disease burden [31]. It indicates that medical factors played consequential roles, which is consistent with calls to "follow the science." But it also suggests that governments learned from each other or, given the short time frame, were possibly increasingly pressured as time passed to issue SAHOs to comply with normative pressures [40]. While there is some evidence that regimes granting greater freedoms to their citizens were less likely to issue a SAHO, the evidence is far from conclusive. Finally, there is no evidence that social or economic factors were consequential despite the important roles often attributed to them in public health research (e.g., [14]). In all, the results suggest that those favoring a public health-centric response to the COVID-19 pandemic should be pleased with the approach of African leaders.

## Author Contributions

**Conceptualization:** Gregg R. Murray, Joshua Rutland.

**Writing – original draft:** Gregg R. Murray, Joshua Rutland.

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
