## [Decision Letter · Decision Letter 0]

27 Sep 2021

PGPH-D-21-00327

Too Much Politics in “Pandemic Politics”? Factors Affecting the Issuance of Stay-at-Home Orders in Response to COVID-19 in Africa

Dear Dr. Murray,

Thank you for submitting your manuscript to PLOS Global Public Health. After careful consideration, we feel that it has merit but does not fully meet PLOS Global Public Health’s publication criteria as it currently stands. Therefore, we invite you to submit a revised version of the manuscript that addresses the points raised during the review process.

We look forward to receiving your revised manuscript.

Kind regards,

Nick Drydakis, Ph.D

Academic Editor

Journal Requirements:

1. Please provide separate figure files in .tif or .eps format only.

2. Please note that your Data Availability Statement is currently missing the repository name and/or the DOI/accession number of each dataset OR a direct link to access each database. If your manuscript is accepted for publication, you will be asked to provide these details on a very short timeline. We therefore suggest that you provide this information now, though we will not hold up the peer review process if you are unable.

Reviewers' comments:

Reviewer's Responses to Questions

**Comments to the Author**

1. Does this manuscript meet PLOS Global Public Health’s publication criteria? Is the manuscript technically sound, and do the data support the conclusions? The manuscript must describe methodologically and ethically rigorous research with conclusions that are appropriately drawn based on the data presented.

Reviewer #1: Partly

2. Has the statistical analysis been performed appropriately and rigorously?

Reviewer #1: Yes

3. Have the authors made all data underlying the findings in their manuscript fully available (please refer to the Data Availability Statement at the start of the manuscript PDF file)?

Reviewer #1: Yes

4. Is the manuscript presented in an intelligible fashion and written in standard English?

Reviewer #1: Yes

5. Review Comments to the Author

Reviewer #1: The manuscript offers a quantitave assessment of the role that five factors related to public health policy (i.e. political, social, economic, medical, and external factors) played in the issuance of stay-at-home orders in response to COVID-19 in Africa.

I find the paper and the arguments discussed of interest and this is a potential contribution, but for a journal like PLOS Global Public Health the paper needs a more strong and consistent theoretical framework within which the nine tested hypothesis are being developed. Thus, authors may relate more clearly the empirical results to the theoretical framework mobilized in developing the hypothesis.

For this reason, I think that this paper needs major revision to be published in this journal.

Despite the rigorous application of the selected methodological frame (event history analysis), the following points should be considered before publication:

Points to be considered:

- Within the introduction the authors argue that the period under observation concerns “the first six months of the pandemic”. However, in the section “METHOD AND DATA” is specified that the period scrutinized in the manuscript “captures every day from January 31 to June 7, 2020”, which is a timespan of less than six months. Then, within the conclusion authors state that: "The event history record only covers about five months, which is an extremely short period of time in policy adoption research". In this respect, authors may introduce the specific (and correct) timespan under study during the introduction, thus providing a rationale that can justify the selected timespan. It does not seem sufficient to say that the observation was interrupted on 7 June, without providing the rationale behind this methodological choice. For which reasons authors decided to stop the observation just 30 days after the last African country issued a SAHO during the first wave of the pandemic? Maybe, the observation period should be extended for increasing the robustness of the results;

- The section “Africa Policy Context” reveals a rather fundamental flaw in the form of an insufficient theoretical frame. I suggest to further deepen the discussion of the current policy context in Africa, with particular attention to socio-political issues concerning public health policy making during outbreaks and pandemics. This can help authors to further clarify their contribution to the current public and academic debate about public health policy making within the Africa region, with particular attention to issues related to the public management of epidemics and pandemics. Although the authors are aware that the African continent is extremely heterogeneous in terms of culture, social and socio-demographic structures, as well as legal and governmental arrangements, in this section we find many “one-size fits all” statements (e.g. “Political parties have been plagued by inexperience and, as a result, political institutions have remained weak in many cases”; “African governments have been crippled for decades by corruption, ineffective policy creation, and terroristic armed militias”). I therefore suggest to rework this section, thus to offer a clearer picture (at least in terms of regional areas or sub-areas) of the current characteristics of the African policy context by mobilizing pertinent academic contributions for understanding policy making in the Africa region;

- Although the hypotheses seems to be pertinent, they should be discussed more carefully in relation to the current debate on public health policymaking, so as to better clarify the contribution offered by this paper and its relevance for the understanding of public health policymaking during pandemics;

- Concerning the hypothesis H3b, I suggest to further analyze the role of socio-demographic variables with respect to public health policymaking processes. It should also be clarified what the concept of “risk” mentioned in the H3b hypothesis refers to. It is not clear in which ways the hypothesis H3b is related to the socio-demographic variables;

- As regards external factors, the authors refer to the “policy diffusion theory”. With the aim to emphasize both geographical and temporal diffusion patterns of policies, the authors (mainly citing Grubler) argue that policies spread like other innovations, such as the canal, railway, telegraph, road, and oil pipeline networks. The adoption of this theoretical perspective should be better justified, especially by explaining how the health policy transition have been interpreted and explained under the lens of the policy diffusion theory. Thus, it could be of interest to clarify to which extent a novel policy (as innovative immaterial artifact) can follow diffusion patterns similar to technological innovation processes.

- Result section: my main concern is whether this section (after a possible review process in which the authors further deepen the theoretical argumentation behind the proposed tested hypothesis) could become substantial enough for PLOS Global Public Health, since as it stands now it seems to be a descriptive research report, with loose connection with relevant academic debates about public health policy making in pandemic times. Indeed, despite presenting a potentially intriguing picture on how five main factors (i.e. political, social, economic, medical, and external factors) affect the issuance of stay-at-home orders in response to COVID-19 in Africa. So, authors should relate their findings to the insights in previous studies on public health policy making in the context of African continent.

6. PLOS authors have the option to publish the peer review history of their article (what does this mean?). If published, this will include your full peer review and any attached files.

**Do you want your identity to be public for this peer review?** For information about this choice, including consent withdrawal, please see our Privacy Policy.

Reviewer #1: No

---

## [Editor Report · Decision Letter 1]

18 Nov 2021

Prioritizing Public Health? Factors Affecting the Issuance of Stay-at-Home Orders in Response to COVID-19 in Africa

PGPH-D-21-00327R1

Dear Dr. Murray,

We're pleased to inform you that your manuscript has been judged scientifically suitable for publication and will be formally accepted for publication once it meets all outstanding technical requirements.

Within one week, you'll receive an e-mail detailing the required amendments. When these have been addressed, you'll receive a formal acceptance letter and your manuscript will be scheduled for publication.

An invoice for payment will follow shortly after the formal acceptance. To ensure an efficient process, please log into Editorial Manager at https://www.editorialmanager.com/pgph/ click the 'Update My Information' link at the top of the page, and double check that your user information is up-to-date. If you have any billing related questions, please contact our Author Billing department directly at authorbilling@plos.org.

Kind regards,

Nick Drydakis, Ph.D

Academic Editor
